# Practice of and associated factors regarding prevention of surgical site infection among nurses working in the surgical units of public hospitals in Addis Ababa city, Ethiopia: A cross-sectional study

Ayelign Mengesha[1]*, Nete Tewfik[2]◍, Zeleke Argaw[2]◍, Biruk Beletew[1‡], Mesfin Wudu[1‡]

1 Department of Nursing, College of Health Sciences, Woldia University, Woldia, Amhara regional state, Ethiopia, 2 Department of Nursing, College of Health Sciences, Addis Ababa University, Addis Ababa, Ethiopia

◍ These authors contributed equally to this work.
‡ These authors also contributed equally to this work.
* Ayelignmengesha59@gmail.com

## Abstract

### Background

Surgical site infections are one of the commonest types of healthcare-associated infections. Up to 60% of these infections are estimated to be preventable by using evidence-based guidelines. As a front line caregiver, nurses are responsible for the majority of preventive activities. Hence, the enhanced practical skill of nurses is an essential component in preventive actions.

### Objective

The aim of this study was to assess the practice of nurses and identify factors associated with it regarding prevention of surgical site infections in Addis Ababa city public hospitals.

### Methods

An institution-based cross-sectional study was carried out from March 01–30, 2018. An adapted and pretested, self-administered questionnaire was utilized as a data collection tool. A stratified random sampling technique was employed by considering the level of hospitals as a stratum. Data were entered into a computer using Epi-data 3.1 statistical package. Then, it was exported to SPSS Version 23 for further analysis. Descriptive statistics were computed for the study variables. Bivariate regression analysis was also run to assess the association between independent variables and the level of nurse's practice. To see the relative effect of independent variables on the nurse's practice, a multivariable regression analysis was carried out.

**Data Availability Statement:** All relevant data are within the paper and its Supporting Information files.

**Funding:** The study was funded by Addis Ababa University. However, the funder had no role in study design, data collection and analysis, decision to publish, or preparation of the manuscript.

**Competing interests:** The authors have declared that no competing interests exist.

## Result

A total of 409 nurses returned the questionnaire with a response rate of 98%. Majority (60.4%) of the participants were females and 84.1% were BSc holders. Less than half, (48.9%) of the participants were found to have good practice regarding prevention of surgical site infection. Being male, having more work experience, higher educational level and using available infection prevention guidelines were significantly associated with practice at p <0.05.

## Conclusion

More than half of the participants have inadequate practice regarding prevention of surgical site infection. Training nurses, making surgical site infection prevention guidelines easily accessible and ensuring possessed knowledge by nurses is potent enough and can be translated into desirable actions are recommended.

## Introduction

Surgical site infections (SSIs) are a serious complication of surgical procedures and the leading healthcare-associated infections (HAIs) in settings with limited resources [1]. HAIs are acquired in health care settings and affect patients, health care workers, and other caregivers as well [1, 2]. Most of SSIs are caused by multidrug-resistant microorganisms [1–3].

Up to 60% of SSIs have been estimated to be preventable by using evidence-based guidelines. Despite this fact, SSIs are still the leading HAIs reported hospital-wide in low and middle-income countries (LMICs) [4].

The majority of SSIs are attributable to the compromised quality of health care due to a shortage of resources being made available by the healthcare settings, patient's characteristics and, the low quality of care delivered in the health care settings [1, 4–6].

Surgical site infections account for 20% of all HAIs in hospitalized patients and each SSI is associated with approximately 7–11 additional postoperative hospital-days. Besides, patients with SSI have a 2–11 times higher risk of death compared with operative patients without SSI, and 77% of deaths in patients with SSI are directly attributable to the infection [1, 6].

According to a 2014 WHO report, the estimated prevalence rate of SSIs within the past two decades ranges to 19.6% in Europe and 20% in the United States of America (USA). Similarly, in Africa, the incidence rate of SSIs is reported ranging from 12% in Algeria to 31% in Nigeria [7].

The rate of SSIs is declining in developed countries. In the USA, a more recent study conducted to describe the epidemiology of complex SSIs on 29 community hospitals revealed that the overall prevalence rate of SSI was 0.7 infections per 100 procedures [8].

The prevalence rate of SSI also significantly varies from region to region and country to country [9]. A systematic review in Iran has reported that SSIs were the third most frequent hospital-acquired infections with an estimated prevalence rate of 4.7–25% in patients undergoing elective colorectal surgeries in various countries [10].

A study in Cameron has revealed that the prevalence rate of SSI was around 9.2% on a study conducted in three hospitals in the country in patients who undergo different types of surgeries in these settings [11]. Another study done in Africa has shown that the cumulative incidence rate of SSIs ranges from 2.5 to 30.9% [12].

Ethiopia shares the burden of SSIs and the infection rate is reported ranging from 10.9%, in Bahirdar to 19.1% in Hawassa respectively [13, 14]. Another study among patients with clinical signs of post-surgical wound infection in Ethiopia, has revealed that the prevalence rate of

culture-confirmed SSI was 75% and isolated bacterias have shown multi-drug resistance to the commonly used antibiotics in the hospital in 82.9% of the patients with SSI [15].

Surgical site infections also pose a huge challenge in terms of additional costs to the health systems and service payers [16]. The estimated hospital-related cost of a single SSI ranges from $12,000–$35,000 [17].

The surgical team can play a crucial role in preventing efforts by following surgical safety checklists in terms of limiting the number of people in the operating room and transit, minimizing conversations at the time of surgery, closing the door, improving the ventilation system, making timely decisions on the type of antimicrobial prophylaxis and adequately preparing the patient's skin and hands of the surgical team, during the pre, intra and post-operative periods [3, 18, 19].

Unlike other health professionals, nurses spent the majority of their time with patients and cover most of SSI prevention activities. This shows that nurses are the primary responsible bodies and can play a central role in preventing efforts by improving the quality of care they deliver, for example; improving the improper use of prophylactic antibiotics, poor hand hygiene practice, improper donning and doffing of personal protective equipment, skin preparation practices and proper implementation of all other surgical safety checklists [3, 19].

However, studies on nurse's practice regarding the prevention of SSIs in Ethiopia are limited. Hence, this study aimed to assess the practice of nurses and identify factors associated with it regarding the prevention of SSIs.

## Methods

### Study setting, design and period

An institution-based cross-sectional study was carried out at Addis Ababa city. According to the 2017 projected estimation, the city has 6.6 million people [20]. The study was conducted at four public hospitals found in the city, from March 01–30, 2018. These hospitals were, Tikur Anbesa specialized and teaching hospital (TASH), Armed forces comprehensive specialized hospital (AFCSH), Yekatit 12 hospital medical college (Y12HMC) and Tirunesh Beijing general hospital (TBGH).

### Study population

Nurses working in the surgical units of selected public hospitals in Addis Ababa city were the study populations.

### Sample size determination

The sample size for this study was determined by using the single population proportion formula considering the assumptions: The proportion of nurses having a good practice regarding prevention of SSIs being 48.7% from a previous study (p = 0.487) [21]. Level of significance 5% ($\alpha$ = 0.05), Z $\alpha$/2 = 1.96 and margin of error 5% (d = 0.05). The sample size was calculated as follows:

$$n_o = \frac{Z\left(\alpha/2\right)^2 * p\left(1-p\right)}{d^2}$$

Adding a 10% non-response rate and multiplying by 1.5 (design effect) the total sample size required for this study appeared to be **417**.

## Sampling technique

A stratified simple random sampling technique was employed. First, the public hospitals found in Addis Ababa city were stratified into secondary and tertiary hospitals, because the public hospitals found in Addis Ababa were secondary and tertiary in level. There were four tertiary and nine secondary hospitals in the town. However, the number of nurses working in tertiary hospitals was very large compared to the secondary hospitals. Therefore, two representative hospitals from each stratum were selected randomly. Then, in these selected hospitals, the number of study participants was assigned proportionally.

## Data collection technique and tools

An adapted and structured, pretested, self-administered questionnaire was employed to collect data from participants. A Likert scale consisting of 25 items with responses being answered in a 4-point scale (never practice, seldom practice, sometimes practice, and always practice), was employed to measure the nurse's practice regarding prevention of SSIs [22]. The scores ranged from 0–75 and were transformed into percentage for interpretation. The mean score was utilized as a cut of point to describe the nurse's level of practice regarding prevention of SSIs.

Two supervisors and four data collectors, who had BSc degree in nursing, were recruited to assist in the data collection process. Training was given for the supervisors and data collectors, three days before the actual data collection period. The training was given on the objectives of the study, the questions, and extent of explanations, the way to keep privacy and maintaining the confidentiality of the information and related ethical issues.

## Data quality assurance

To ensure the quality of data, a structured and pretested self-administered English version questionnaire was utilized because English is an academic language in Ethiopia. The questionnaire was pretested on 5% of the sample size by taking 21 nurses. The pretest study was conducted two weeks before the actual data collection period at Zewditu memorial hospital. The hospital was selected by lottery method among the hospitals found in the city which weren't included in the study settings. The study subjects who fulfill the inclusion criteria were selected by simple random sampling technique.

The validity of the questionnaire was done in a previous study. In that study, it was verified by 5 experts in the surgical and infection control fields, yielding a content validity index of 0.98. In the reliability test, a Cronbach Alpha coefficient of 0.91was obtained [22]. In the present study, the Cronbach's Alpha coefficient appeared to be 0.89. Besides, all the necessary amendments were also done on the instructions of the questionnaire, contents, order of the questions, and grammatical issues based on the pretest findings. All data were checked for completeness, accuracy, clarity, and consistency by the supervisors and the principal investigator immediately after the data were collected. Double data entry and validation were performed and the data were intensively cleaned before analysis.

## Data processing and analysis

The data were coded and entered into a computer using Epi-data 3.1 Statistical program and were exported to SPSS Version 23 for further analysis. Then, descriptive statistics were computed for the study variables. The mean score was used to categorize the nurse's practice level as good and poor. Those participants who answered mean and above the mean score of practice questions were categorized as having a good practice regarding prevention of surgical site infection and poor otherwise. Bivariate logistic regression analysis was run to assess the

association between independent variables and the level of nurses practice. Multivariate logistic regression analysis was also carried out to see the relative effects of independent variables on the outcome variable. The odds ratios were calculated to determine the strength of the association between independent variables and the outcome variable, and a 95% confidence interval was utilized to guide the interpretation of results. A P-value of less than 0.05 was taken as a cutoff point to declare a statistically significant association between independent and dependent variables. Finally, the result is presented using texts and tables.

## Ethical approval and consent to participate

Ethical clearance was obtained from Addis Ababa University institutional review board and Addis Ababa city health bureau ethical review committee. Then, the ethical clearance & support letter was taken to the selected hospitals to obtain permission and cooperation during the data collection process. Informed verbal consent was obtained from every study participants after a detailed explanation of the purpose and benefit of the study right before data collection and participation was voluntary and each participant signed on a statement of informed consent after he/she received the questionnaire to be filled. Confidentiality of the information was assured by making the questionnaires anonymous and privacy of the respondents was maintained.

## Results

### Sociodemographic characteristics of respondents

A total of four hundred nine nurses (409), 277 (67.7%) from tertiary hospitals (TASH and AFCSH) and 132 (32.3%) from secondary hospitals (Y12HMC and TBGH)) completed and returned the questionnaire making the response rate 98%. Two hundred seventy-four (60.4%) of them were females. The mean age score was 31.16 and the median was 30 years. The minimum age of study participants was 22 years and the maximum was 58 years old. Among the study participants, 221 (54%) were married, 179 (43.8%) single and the remaining 9 (2.2%) were widowed. Most of the participants (84.1%) were BSc holders followed by (10%) masters and (5.9%) diploma nurses. The average monthly income of participants was 5258.94 Ethiopian birr. Regarding work experience, 45.5% of the participants had more than 5 years of total work experience in health care settings. Besides, participants had a minimum of 1 year and a maximum of 22 years of experience in the surgical units. From the total study participants, 224 (54.8%) claim they have taken training regarding infection control methods (Table 1).

### Nurse's practice regarding prevention of SSI

The mean practice score of the study participants was found being 58.36 and the standard deviation was 7.36 with a minimum score of 27 and a maximum score of 75 out of the 25 Likert items. In this study, around half, 200 (48.9%) of the participants had a good practice regarding prevention of SSIs. Concerning the items, of the participants, 259 (63.3%) said, they always use alcohol and chlorhexidine gluconate in their surgical site infection prevention practice. In line with this, 268 (65.5%) replied, they always wash their hands before and after changing wound dressings for a question how often do you wash your hands before and after changing wound dressings (Table 2).

### Factors associated with nurse's practice regarding prevention of SSI

In the bivariate regression analysis participant's age, income, total work experience, work experience in surgical units, taking training on IP methods and using available IP guidelines

**Table 1. Sociodemographic characteristics of nurses in Addis Ababa city public hospitals, 2018 ($n$ = 409).**

| Characteristics | | Frequency | Percentage |
|---|---|---|---|
| Age | < 30years | 238 | 58.2 |
| | ≥ 30years | 171 | 41.8 |
| Sex | Male | 162 | 39.6 |
| | Female | 247 | 60.4 |
| Marital status | Single | 179 | 43.8 |
| | Married | 221 | 54.0 |
| | Divorced | 9 | 2.2 |
| Educational status | Diploma | 24 | 5.9 |
| | BSc degree | 344 | 84.1 |
| | Master's degree | 41 | 10.0 |
| Monthly income | <5258.94 ETB | 223 | 54.5 |
| | ≥5258.94 ETB | 186 | 45.5 |
| Total work experience | Five years or less | 223 | 54.5 |
| | More than 5 years | 186 | 45.5 |
| Experience in surgical units | Three years or less | 301 | 73.6 |
| | More than three years | 108 | 26.4 |
| Duty ward | Surgical | 152 | 37.2 |
| | Orthopedics | 94 | 23.0 |
| | Recovery | 50 | 12.2 |
| | Gyn. and labor | 79 | 19.3 |
| | Others (e.g., Eye, ENT) | 34 | 8.3 |
| Ever took IP training | Yes | 224 | 54.8 |
| | No | 185 | 45.2 |
| Number of IP training's attended | Only once | 170 | 75.9 |
| | More than once | 54 | 24.1 |
| Availability of IP guidelines | Yes | 225 | 55.0 |
| | No | 184 | 45.0 |
| Usage of IP guidelines | Yes | 209 | 51.1 |
| | No | 200 | 48.9 |

ETB—Ethiopian Birr

IP—Infection Prevention

were significantly associated with the practice of nurses regarding prevention of SSIs. However, on multivariate analysis, only educational status, work experience and using available IP guidelines were significantly associated. The odds of nurses with BSc degree was more than 4 times higher to have a good practice regarding prevention of SSI than diploma nurses (AOR = 4.35, 1.73–10.95) and the odds of those who use available IP guidelines in their routine practice was about 3 times higher to have good practice than those who did not (AOR = 2.72, 1.73–4.28). Similarly, the odds of nurses with work experience of more than 5 years was about 2 times (AOR = 1.71, 1.10–2.64) higher to have a good practice regarding prevention of SSI than those who have 5 years or less experience (Table 3).

Nurses were asked to rate their level of practice, and 56 (13.7%) of them rated their practice as unsatisfactory. Then, they were asked to mention the possible factors affecting their practice and they put their responses as follows: I have no sufficient knowledge about SSIs, 19 (4.6%), inadequate resources to implement surgical safety checklists, 154 (37.7%), insufficient performance monitoring systems, 101 (24.7%), lack of SSI assessment and preventive measure

Table 2. Nures's response to each practice item in Addis Ababa city public hospitals, 2018 (*n* = 409).

| SSI prevention practice items | Frequency and percentage | | | |
|---|---|---|---|---|
| | Never | Seldom | Sometimes | Always |
| Use alcohol and chlorhexidine gluconate | 2(0.5) | 6 (1.5) | 142(34.7) | 259(63.3) |
| Wash hands before and after changing wound dressings. | 2(0.5) | 21(5.1) | 118(28.9) | 268(65.5) |
| Wash hands before wearing surgical gloves. | 2(5.6) | 61(14.9) | 194(47.4) | 131(32) |
| Perform preoperative shaving on the day before surgery. | 213(52.1) | 74(18.1) | 76(18.6) | 46(11.2) |
| Learn shaving methods from others. | 124(30.3) | 93(22.7) | 129(31.5) | 63(15.4) |
| Administer preoperative prophylactic antibiotics within 120 minutes before surgery. | 37(9.0) | 39(9.5) | 113(27.6) | 220(53.8) |
| Advise patients to take pre-operative showering within 6–12 hours before surgery. | 43(10.5) | 56(13.7) | 179(43.8) | 131(32) |
| Advise patients to take pre-operative showering with an antimicrobial agent. | 43(10.5) | 57(13.9) | 186(45.5) | 123(30.1) |
| Perform prescribed glucose tests before and after surgery in a diabetic patient. | 6(1.5) | 16(3.9) | 67(16.4) | 320(78.0) |
| Assess the patient's body mass index before and after surgery. | 88(21.5) | 116(28.2%) | 146(35.7) | 59(14.4) |
| Administer injection insulin or oral medication as ordered in diabetic patients. | 8(2.0) | 4(1%) | 76(18.6) | 321(78.5) |
| Advise obese patients to reduce the amount of carbohydrates they take. | 31(7.6) | 54(13.2%) | 196(47.9) | 128(31.3) |
| Advise a malnourished patient to take a nutritious diet. | 1(0.2) | 17(4.2) | 133(32.5) | 258(63.1) |
| Advise immuno-compromised patients to avoid contact with people having infections. | 6(1.5) | 32(7.8) | 151(36.9) | 220(53.8) |
| Advise a malnourished patient to take vegetables and fruits before & after surgery. | 10(2.4) | 22(5.4) | 120(29.3%) | 257(62.8) |
| Use sterile dressing materials for cleaning a surgical wound. | 2(0.5) | 9(2.2) | 76(18.6) | 322(78.7) |
| Advice patients with immunodeficiency disorder to maintain their hygiene. | 2(0.5) | 7(1.7%) | 71(17.4%) | 329(80.4) |
| Follow aseptic technique to obtain swab culture. | 2(0.5) | 22(5.4) | 62(15.2) | 323(79.0) |
| Follow aseptic technique during dressing. | 1(0.2) | 6(1.5) | 89(21.8) | 313(76.5) |
| Use povidone-iodine and normal saline for cleansing surgical wound dressing. | 2(0.5) | 7(1.7) | 146(35.7) | 254(62.1) |
| Asses and monitor surgical site conditions. | 3(0.7) | 6(1.5) | 99(24.2) | 301(73.6) |
| Separate infected dressing from non-infected. | 0(0.0) | 19(4.6) | 138(33.7) | 252(61.6) |
| Wear a face mask during surgical wound care. | 2(0.5) | 65(15.9) | 273(66.7) | 69(16.9) |
| Clean and disinfect the surface of the dressing trolley with anti-septic solutions. | 0(0.0) | 9(2.2) | 186(45.0) | 214(52.3) |
| Discard the soiled materials in the proper place after performing wound dressing. | 0(0.0) | 2(0.5) | 48(11.7) | 359(87.8) |

feedback systems, 60 (14.7%). And other reasons, 372 (91%) like; excessive workload, staff inadequacy, lack of training to upgrade their level of practice, small chance to learn and develop knowledge and skills through formal education, lack of encouraging programs, insufficient orientation programs during unit rotation, unsuitable hospital environment and negligence and lack of interest as a result of the harassing hospital environment were mentioned as factors affecting their level of practice (Table 4).

## Discussion

Surgical site infections are one of the serious complications of surgical procedures and the most common type of healthcare-associated infections (HAIs) [1]. Up to 60% of SSIs have been estimated to be preventable by using evidence-based guidelines [1, 5, 6]. It takes a multi-disciplinary approach to prevent SSIs. As a frontline caregiver, nurses can help surgical patients avoid SSI through pre and intra-operative implementation of surgical safety checklists and adequate post-operative wound care and thorough discharge planning [2, 5, 6]. Therefore, this study aimed to assess the practice of nurses and identify factors associated with it regarding prevention of SSIs.

This study examine the level of nurse's practice regarding prevention of SSIs, and less than half (48.9%) of them were found having a good practice regarding prevention of SSI. This means, in the reverse more than half of the nurses were practicing poorly regarding prevention

**Table 3. Logistic regression analysis of factors associated with the practice of nurses regarding prevention of SSI in Addis Ababa city public hospitals, Ethiopia, 2018 (*n* = 409).**

| List of independent Variables | Practice of nurse's | | Bivariate logistic regression | Multivariate logistic regression |
|---|---|---|---|---|
| | Good | Poor | COR (95%CI) | AOR (95%CI) |
| Sex | | | | |
| Male | 84(51.9%) | 78(48.1%) | 1.22(0.82–1.81) | 1.22(0.78–1.92) |
| Female | 116(47.0%) | 131(53.0%) | | |
| Age | | | | |
| ≥ 30 years | 94(55.0%) | 77(45.0%) | 1.52(1.02–2.26)* | 0.84(0.46–1.51) |
| < 30 years | 106(44.5%) | 132(55.5%) | | |
| Educational status | | | | |
| Diploma | 8(33.3%) | 16(66.7%) | | |
| Degree | 171(49.7%) | 173(50.3%) | 1.98(0.82–4.74) | **4.35(1.73–10.95)*** |
| Masters | 21(51.2%) | 20(48.8%) | 2.1(0.74–5.98) | 2.20(0.75–6.46) |
| Level of hospital | | | | |
| Secondary | 68(51.5%) | 64(48.5%) | | |
| Tertiary | 141(50.9%) | 136(49.1%) | 1.03(0.68–1.55) | 1.09(0.70–1.71) |
| Monthly income | | | | |
| ≥ 5258.94 ETB | 105(56.5%) | 81(43.5%) | 1.75(1.18–2.59)* | 1.01(0.56–1.84) |
| < 5258.94 ETB | 95(42.6%) | 128(57.4%) | | |
| Total work experience | | | | |
| More than five years | 105(56.5%) | 81(43.5%) | 1.75 (1.18–2.59)* | **1.71(1.10–2.64)*** |
| Five years or less | 95(42.6%) | 128(57.4%) | | |
| Experience in surgical unit | | | | |
| More than three years | 64(59.3%) | 44(40.7%) | 1.77(1.13–2.76)* | 1.33(0.74–2.40) |
| Three years or less | 136(45.2%) | 165(54.8%) | | |
| Ever took IP training | | | | |
| Yes | 132(58.9%) | 92(41.1%) | 2.01(1.35–2.99)* | 1.90(0.16–3.11) |
| No | 68(36.8%) | 117(63.2%) | | |
| No of IP training's attended | | | | |
| More than once | 35(62.5%) | 21(37.5%) | 1.29(0.69–2.41) | 1.21(0.59–2.47) |
| Only once | 99(57.1%) | 73(42.4%) | | |
| Usage of IP guidelines | | | | |
| Yes | 125(57.6%) | 84(40.2%) | 2.63(1.76–3.94)* | **2.72 (1.73–4.28)*** |
| No | 75(37.5%) | 125(62.5%) | | |

* Significant at p<0.05.

of SSI. This finding is in agreement with studies conducted in Bangladesh, Nigeria, Tanzania, and Ethiopia (Amhara regional state), in which the level of nurse's practice towards prevention of SSIs was at a low level [23–26]. These studies suggested that the practice of nurses regarding prevention of SSI was affected by multiple factors giving lack of training on SSI prevention methods in line with the latest global and national guidelines with the latest recommendations as an example.

However, the finding is in contrast with a study in Pakistan and another study in Bangladesh in that the overall practice of staff nurses regarding preventing and managing surgical site infection was at a good level [27, 28]. This difference might be due to differences in sample size, attitude, training and workload of nurses regarding prevention of SSI. It might also be due to differences in the developmental level of the countries and the resulting shortage of

**Table 4. Nurse's self-rating of current practice regarding prevention of SSI in Addis Ababa city public hospitals, Ethiopia, 2018 (*n* = 409).**

| Questions | Responses | Frequency | Percentage |
|---|---|---|---|
| How did you rate the overall level of your current practice regarding prevention of SSI? | Very unsatisfactory | 1 | 0.2% |
| | Unsatisfactory | 55 | 13.4% |
| | Satisfactory | 341 | 83.4% |
| | Very satisfactory | 12 | 2.9% |
| If you are not very satisfied with your current level of practice, what are the reasons/factors? | I have no sufficient knowledge about SSIs | 19 | 4.6% |
| | Inadequate resources to implement surgical safety checklists | 154 | 37.7% |
| | Insufficient performance monitoring systems | 101 | 24.7% |
| | Lack of surgical site infection assessment and preventive measure feedback systems | 60 | 14.7% |
| | Others | 372 | 91% |

resources as nurses reported that lack of resources to implement SSI prevention activities was one of the major factors affecting their practice regarding prevention of SSIs.

On the multivariate analysis, higher educational status, more work experience and using available IP guidelines were significantly associated with the practice of nurses. The odds of nurses with BSc degree was about 4 folds higher to have a good practice regarding prevention of SSI than diploma nurses and for those who use available IP guidelines in their routine practice, the odds ratio was about 3 folds higher to have a good practice than those who did not. The possible reasons for these findings might be because the majority of participants were BSc degree holders and diploma nurses were very small in number and again those who claim they are using available IP guidelines in their routine practice were larger in number than diploma nurses. The other reason might be the number of credit hours learned and the duration of college and university stay which is different between them. This finding is consistent with a study in Tanzania in which undergraduate nurses demonstrated and scored higher in their practice compared to diploma and certificate nurses (P = 0.003) [25].

However, it is in contrast with a study in Ethiopia, Amhara regional state in which diploma nurses scored better in practice compared to BSc nurses [26]. These differences might be due to differences in, workload, training, and attitude of nurses regarding prevention of SSIs. It may be also due to the assessment tool which is vulnerable to the central tendency, acquiescence and social desirability biases [28, 29]. The odds ratio of nurses with work experience of more than 5 years was about 2 folds higher to have a good practice regarding prevention of SSI than those who have 5 years or less experience. The finding is in line with other studies in which work experience was significantly associated with the nurse's level of practice [25, 26]. The possible explanation is, as the nurse's service year increases, they are more likely to be exposed to work with experienced staff and acquire knowledge and necessary skills through the process.

After completing the Likert scale nurses were asked to rate their overall level of practice regarding prevention of SSI, and the result revealed that fifty-six (13.7%) of the participants rated their level of practice as unsatisfactory which is highly significant in number and effect on the rate of SSI and its prevention. Then, they were asked to mention the possible factors affecting their level of practice and insufficient knowledge about SSIs, inadequate resources to implement surgical safety checklists, insufficient performance monitoring systems, lack of SSI assessment and preventive measure feedback systems, and others, like; excessive workload, staff inadequacy, lack of orientation programs during unit rotation, and harassing hospital environment were mentioned as an important factor. This is highly worrying, these factors may result in feelings of frustration among nurses and may end up with professional non-achievement for nurses and low-quality care to the health systems. The finding is consistent

with a Bangladesh study in which, insufficient knowledge, inadequate resources and budgets, insufficient performance monitoring systems, and lack of surveillance systems were identified as major factors affecting the nurse's level of practice in SSI prevention efforts [23].

## Conclusion

The result of this study revealed that more than half of the participants had a poor level of practice. Higher Educational levels, more experience and using available IP guidelines were significantly associated with the practice of nurses regarding prevention of SSI. Insufficient knowledge, inadequate resources to implement surgical safety checklists, insufficient performance monitoring systems, lack of surgical site infection assessment and preventive measure feedback systems, and others, like; excessive workload, staff inadequacy, lack of training, insufficient orientation programs during unit rotation and harassing hospital environment were also identified as factors affecting the nurse's practice regarding prevention of SSIs. Therefore, attention should be given on upgrading nurse's practice; through training, making resources (e.g. IP prevention guidelines) available and establishing an infection prevention committee in health institutions/ hospitals that works on and communicates with staff nurses, and shares contemporary practices regarding SSI prevention. Furthermore, hospital administrators should emphasis on involving nurses in designing strategies so that nurses can be encouraged and work effectively.

## Limitation

Since this study had used a self-administered Likert scale to measure the nurse's practice regarding prevention of surgical site infections, the following limitations were inherent: avoidance of using extreme response categories by participants—central tendency bias, agreeing with statements as presented—acquiescence bias and participants attempt to portray themselves or their organization in a more favorable way—social desirability bias [28, 29].

## Supporting information

**S1 File.**
(SAV)

## Acknowledgments

The authors would like to express our deepest gratitude to the data collectors and supervisors who helped us in monitoring and supervising the data collection process. Our deepest gratitude also goes to those who participated in this study.

## Author Contributions

**Conceptualization:** Ayelign Mengesha, Nete Tewfik, Zeleke Argaw, Biruk Beletew, Mesfin Wudu.

**Data curation:** Ayelign Mengesha, Nete Tewfik.

**Formal analysis:** Ayelign Mengesha, Nete Tewfik, Zeleke Argaw, Biruk Beletew, Mesfin Wudu.

**Funding acquisition:** Ayelign Mengesha.

**Investigation:** Ayelign Mengesha, Nete Tewfik, Zeleke Argaw, Biruk Beletew, Mesfin Wudu.

**Methodology:** Ayelign Mengesha, Nete Tewfik, Zeleke Argaw, Biruk Beletew, Mesfin Wudu.

**Project administration:** Ayelign Mengesha.

**Supervision:** Ayelign Mengesha, Nete Tewfik.

**Validation:** Ayelign Mengesha, Nete Tewfik, Zeleke Argaw, Biruk Beletew, Mesfin Wudu.

**Visualization:** Ayelign Mengesha, Nete Tewfik, Zeleke Argaw, Biruk Beletew, Mesfin Wudu.

**Writing – original draft:** Ayelign Mengesha, Nete Tewfik, Zeleke Argaw, Biruk Beletew, Mesfin Wudu.

**Writing – review & editing:** Ayelign Mengesha, Nete Tewfik, Zeleke Argaw, Biruk Beletew, Mesfin Wudu.

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
