## [Decision Letter · Decision Letter 0]

10 Mar 2020

PONE-D-19-32890

Nurses practice and associated factors regarding prevention of surgical site infection in Ethiopian hospitals, 2018

PLOS ONE

Dear Mr Kassie,

Thank you for submitting your manuscript to PLOS ONE. After careful consideration, we feel that it has merit but does not fully meet PLOS ONE’s publication criteria as it currently stands. Therefore, we invite you to submit a revised version of the manuscript that addresses the points raised during the review process. The revisions suggested by the reviewers are appropriate and will assist in tightening up the submitted article. I would like to particularly draw your attention to the comments made about the introduction section. 

We would appreciate receiving your revised manuscript by Apr 24 2020 11:59PM. To enhance the reproducibility of your results, we recommend that if applicable you deposit your laboratory protocols in protocols.io, where a protocol can be assigned its own identifier (DOI) such that it can be cited independently in the future. For instructions see: http://journals.plos.org/plosone/s/submission-guidelines#loc-laboratory-protocols

We look forward to receiving your revised manuscript.

Kind regards,

Holly Seale

Academic Editor

PLOS ONE

Both of the reviewers have identified minor suggested changes to improve the flow and direction of your paper

Journal Requirements:

2. Your ethics statement must appear in the Methods section of your manuscript. If your ethics statement is written in any section besides the Methods, please move it to the Methods section and delete it from any other section. Please also ensure that your ethics statement is included in your manuscript, as the ethics section of your online submission will not be published alongside your manuscript.

Reviewers' comments:

Reviewer's Responses to Questions

**Comments to the Author**

1. Is the manuscript technically sound, and do the data support the conclusions?

Reviewer #1: Yes

Reviewer #2: Yes

2. Has the statistical analysis been performed appropriately and rigorously? 

Reviewer #1: Yes

Reviewer #2: Yes

3. Have the authors made all data underlying the findings in their manuscript fully available?

Reviewer #1: Yes

Reviewer #2: No

4. Is the manuscript presented in an intelligible fashion and written in standard English?

Reviewer #1: Yes

Reviewer #2: Yes

5. Review Comments to the Author

Reviewer #1: The topic is interesting and well written however requires some minor changes.

1) Introduction:

i) Paragraph 1 Line 2: ' HAIs are acquired by patients.....'. HAI can be acquired by healthworkers and other care givers as

well and not only by the patients.

ii) Paragraph 2 Line 3: 'Most of these infections are attributable to the poor quality of care delivered by health

workers....' . This is portraying healthworkers in bad light.

Compromised quality of health care also depends on what resources are being made available by the healthcare setting

to the HCWs, patients and care givers. Delivery of healthcare is not the sole responsibility of the healthcare worker.

iii) Paragraph 4,5,6 : The focus of the paper is on a develping country, however in the introduction section more focus is

given on the status of SSI in the develped economies (US, Europe, Middle East) rather than on developing economies

especially other regional economies (countries) . Case should be made by preseting equal amount of information of the

developing world and developed world, if not less.

iv) Last paragraph, Line 4: In addtion to 'improving

the improper use of prophylactic antibiotics, poor hand hygiene' Donning and doffing of PPE is equally important. This

should be added.

2) Materials and Methods : Ok

3) Results:

Table 1 :

Age: Minimum and Maximum age should be included (under characteristics column only). This will help the reader to

better understand the range of workers involved in the surgical sites.

Total Work experience & Experience in surgical units: Minimum and Maximum experience should be included (under

characteristics column only).

Experience in surgical units : Typo error 'More three years' should be ' More than three years'

4) Discussion:

Paragraph 2 Line 2 : 'However, the finding is in contrast with other studies....' . The reader would like to know more

about the studies. More so, because the authors quote that their findings are in contrast to the findings of these

studies.

5) Conclusion:

Line 1: 'The result of this study revealed that more than half of the participants were practicing poorly'. Practicing what

poorly? This may confuse the reader. Elaborate the sentence.

Reviewer #2: Nurses practice and associated factors regarding prevention of surgical site infection in Ethiopian hospitals, 2018

Comment1: In “ introduction part” authors mentioned “Surgical site infection is the most common type of healthcare-associated infections” however it would be better to indicate “…surgical site infection is the leading infection in settings with limited resources” according to WHO Health care-associated infections FACT SHEET, since UTI is most common type of those in developed countries.

Comment2: In “Sampling Technique” part it would be better to mention the number of hospitals in each strata, so that the authors decided to select two from each randomly.

Comment3: In “Data Quality Assurance” part it is better to clarify the pretest study process, whether the pretest was done only for calculating Cronbach’s Alpha or not? “The pretest study was conducted two weeks before the actual data collection period at Zewditu memorial hospital”

Comment4: In “Data Quality Assurance” part it might be more acceptable to give more details about the mentioned action “The necessary amendments were made on; the instructions, contents, order and grammatical issues”

Comment4: In “Data Quality Assurance” part in the article that is referred to “Nurses’ Surgical Site Infection Prevention Practices in Bangladesh” the questionnaire was translated from English to Bangla and then was used for collecting data, therefore it would be preferable that the authors declare which version was used in the current study.

Comment5: In “Nurse’s practice regarding prevention of SSI” part the meaning of percentage (77.8%) in the following sentence is unclear” The mean practice score of the study participants was found being 58.36 (77.8%)”

Comment6: In “Factors associated with nurse’s practice regarding prevention of SSI” part there is one inconsistency between table 3 results and manuscript text about AOR.

"the odds of nurses with work experience of more than 5 years was about 2 times higher to have good practice regarding prevention of SSI than those who have 5 years or less experience (AOR = 2.89, 1.60 - 5.21)"

Comment7: In table 3 the significance level of Bivariate logistic regression results were not shown by” *Not significant, ** Significant at p<0.05.” symbols.

6. PLOS authors have the option to publish the peer review history of their article (what does this mean?). If published, this will include your full peer review and any attached files.

Reviewer #1: Yes: Mohammed Owais Qureshi

Reviewer #2: No

---

## [Author Response · Author response to Decision Letter 0]

17 Mar 2020

Response to Reviewers/editor 

Dear reviewer(s)/editor 

Greetings, 

First of all, I thank you very much on behalf of the co-authors for the very detail, genuine and constructive comments raised by the reviewers and the editor which I believe has improved the manuscript significantly. The modification and, or corrections made during the revision are clarified point by point below. 

 Yours sincerely,

Point to point response on the details of reviewer(s)/editor comments;

Journal Requirements:

1. Please ensure that your manuscript meets PLOS ONE's style requirements, including those for file naming. The PLOS ONE style templates can be found at http://www.plosone.org/attachments/PLOSOne_formatting_sample_main_body.pdf and http://www.plosone.org/attachments/PLOSOne_formatting_sample_title_authors_affiliations.pdf.

Author’s response: Downloaded and the manuscript is adjusted with the journals requirement. 

2. Your ethics statement must appear in the Methods section of your manuscript. If your ethics statement is written in any section besides the Methods, please move it to the Methods section and delete it from any other section. Please also ensure that your ethics statement is included in your manuscript, as the ethics section of your online submission will not be published alongside your manuscript. 

Author’s response: The ethics statement is moved to the last section of methods.

Reviewers' comments: 

Reviewer #1: The topic is interesting and well written however requires some minor changes. Author’s response: Thank you very much for your genuine and constructive comments, we believe so. We have made the necessary corrections.

1) Introduction: 

i) Paragraph 1 Line 2: ' HAIs are acquired by patients.....'. HAI can be acquired by health workers and other care givers as well and not only by the patients. Author’s response: Of course yes, thank you,

---

## [Editor Report · Decision Letter 1]

20 Mar 2020

Practice of and associated factors regarding prevention of surgical site infection among nurses working in the surgical units of public hospitals in Addis Ababa city, Ethiopia: A cross-sectional study

PONE-D-19-32890R1

Dear Dr. Kassie,

We are pleased to inform you that your manuscript has been judged scientifically suitable for publication and will be formally accepted for publication once it complies with all outstanding technical requirements.

With kind regards,

Holly Seale

Academic Editor

PLOS ONE

---

## [Editor Report · Acceptance letter]

30 Mar 2020

PONE-D-19-32890R1 

Practice of and associated factors regarding prevention of surgical site infection among nurses working in the surgical units of public hospitals in Addis Ababa city, Ethiopia: A cross-sectional study 

Dear Dr. Mengesha:

I am pleased to inform you that your manuscript has been deemed suitable for publication in PLOS ONE. Congratulations! Your manuscript is now with our production department. 

With kind regards,

on behalf of

Dr. Holly Seale 

Academic Editor

PLOS ONE